# Optical and Material Characteristics of MoS_2_/Cu_2_O Sensor for Detection of Lung Cancer Cell Types in Hydroplegia

**DOI:** 10.3390/ijms23094745

**Published:** 2022-04-25

**Authors:** Arvind Mukundan, Shih-Wei Feng, Yu-Hsin Weng, Yu-Ming Tsao, Sofya B. Artemkina, Vladimir E. Fedorov, Yen-Sheng Lin, Yu-Cheng Huang, Hsiang-Chen Wang

**Affiliations:** 1Department of Mechanical Engineering, Advanced Institute of Manufacturing with High Tech Innovations (AIM-HI), Center for Innovative Research on Aging Society (CIRAS), National Chung Cheng University, 168, University Rd., Min Hsiung, Chia Yi 62102, Taiwan; d09420003@ccu.edu.tw (A.M.); soul_loving0629@yahoo.com.tw (Y.-H.W.); d09420002@ccu.edu.tw (Y.-M.T.); 2Department of Applied Physics, National University of Kaohsiung, 700 Kaohsiung University Rd., Nanzih District, Kaohsiung 81148, Taiwan; swfeng@nuk.edu.tw; 3Nikolaev Institute of Inorganic Chemistry, Siberian Branch of Russian Academy of Sciences, 630090 Novosibirsk, Russia; artem@niic.nsc.ru (S.B.A.); fed@niic.nsc.ru (V.E.F.); 4Department of Natural Sciences, Novosibirsk State University, 1, Pirogova str., 630090 Novosibirsk, Russia; 5Department of Electronic Engineering, I-Shou University, No. 1, Sec. 1, Syuecheng Rd., Dashu District, Kaohsiung City 84001, Taiwan; yslin@isu.edu.tw; 6Department of Dentistry, Kaohsiung Armed Forces General Hospital, 2, Zhongzheng 1st. Rd., Kaohsiung City 80284, Taiwan

**Keywords:** photoelectrochemical, chemical vapor deposition, molybdenum disulfide (MoS_2_), cuprous oxide (Cu_2_O), positive oxide trap state, DNA, biosensor

## Abstract

In this study, n-type MoS_2_ monolayer flakes are grown through chemical vapor deposition (CVD), and a p-type Cu_2_O thin film is grown via electrochemical deposition. The crystal structure of the grown MoS_2_ flakes is analyzed through transmission electron microscopy. The monolayer structure of the MoS_2_ flakes is verified with Raman spectroscopy, multiphoton excitation microscopy, atomic force microscopy, and photoluminescence (PL) measurements. After the preliminary processing of the grown MoS_2_ flakes, the sample is then transferred onto a Cu_2_O thin film to complete a p-n heterogeneous structure. Data are confirmed via scanning electron microscopy, SHG, and Raman mapping measurements. The luminous energy gap between the two materials is examined through PL measurements. Results reveal that the thickness of the single-layer MoS_2_ film is 0.7 nm. PL mapping shows a micro signal generated at the 627 nm wavelength, which belongs to the B2 excitons of MoS_2_ and tends to increase gradually when it approaches 670 nm. Finally, the biosensor is used to detect lung cancer cell types in hydroplegia significantly reducing the current busy procedures and longer waiting time for detection. The results suggest that the fabricated sensor is highly sensitive to the change in the photocurrent with the number of each cell, the linear regression of the three cell types is as high as 99%. By measuring the slope of the photocurrent, we can identify the type of cells and the number of cells.

## 1. Introduction

Molybdenum disulfide (MoS_2_) is one of the two-dimensional (2D) transition metal chalcogenides that have been studied recently [1,2,3,4,5,6]. Materials related to nanometer-scale electronic and optoelectronic components, such as field-effect transistors, prospective memory components, light-emitting diodes (LED), and sensors, have been manufactured because of the excellent spin-valley coupling and flexural and optoelectronic properties of MoS_2_ [7,8,9,10,11,12,13,14,15,16,17,18,19]. However, MoS_2_ is a layered structure, which has good lubricity, resistance to pressure, and wear resistance, and is mostly used as solid lubricant [20]. It is used in equipment operating under high-speed, heavy-load, high-temperature, and chemical corrosive conditions. MoS_2_ is a black hexagonal crystal structure with silver-gray luster, its Moiré mass is 160.07 g/mol, density is 5.06 g/cm^3^, and melting point is 1185 °C. MoS_2_ itself is insoluble in water, generally insoluble in other acids, alkalis, and organic solvents. However, at 400 °C, oxidation occurs slowly, and molybdenum trioxide (MoO_3_) gradually forms. PN heterostructures have been widely used in the semiconductor industry; as a typical P-type semiconductor, Cu_2_O is considered to be the most effective material [21]. With these characteristics, Cu_2_O can be effectively used as an absorption layer in the visible wavelength range [22,23,24,25]. In a bulk MoS_2_ semiconductor, an indirect energy gap of 1.2 eV between the G- and S-point conduction bands, and a single-layer MoS_2_ semiconductor has a direct energy gap of 1.84 eV. Studies on light absorption, light reflection, and light excitation spectroscopy have evolved from bulk materials to single-layer TMDs [26]. In 2014, Yu-Fei Zhao et al. used a simple chemical synthesis method to cover MoS_2_ successfully on p-Cu_2_O semiconductors for high-efficiency solar hydrogen production [21]. In 2016, Xinne Zhao et al. assembled Cu_2_O nanoparticles and MoS_2_ nanosheets with a 2D planar structure into a 3D MoS_2_/Cu_2_O porous nanocomposite through hydrothermal synthesis [27,28]. In 2017, Linxia Fang et al. successfully decorated Cu_2_O nanoparticles on flower-like MoS_2_ and used them for non-enzymatic current detection in glucose. This structure has Cu_2_O nanoparticles dispersed in MoS_2_ [29]. Gang Li et al. reported the hydrothermal synthesis of MoS_2_/Cu_2_O nanocomposites with a tunable heterojunction to enhance the photochemical activity and stability of visible light [30]. The lack of energy gap of graphene in two-dimensional materials has shifted the focus on two-dimensional transition metal dichalcogenides (TMDs) with different energy gaps obtained by changing the number of layers. In the review of previous studies on MoS_2_/Cu_2_O, MoS_2_/Cu_2_O has high-performance photocatalysis, excellent electrocatalysis, strong stability, reproducibility, and high selectivity for non-MEI sensors. This structure can be used as a hydrogen oxide biosensor for environmental engineering applications, such as energy storage and water purification. It can also be used as a non-MEI current detection in glucose oxidation. It is better for the signal interference of uric acid, dopamine, and ascorbic acid. Apart from inorganic biosensors, organic materials are also getting attention in the last few years. Some biosensors using poly(methylene blue) (PMB), poly(alizarin yellow R), poly(azure A), poly(azure B), poly(azure C), poly(brilliant cresyl blue), and poly(thionine) has been widely studied and reported [31,32,33,34,35,36,37]. Even though these biosensors have good selectivity, their disadvantages include instability, cost, and a need for a mediator in some cases [38]. On the other hand, in recent years due to the excellent electrochemical properties of Cu_2_O, much research has been conducted towards its application as a sensor [39,40,41,42]. MoS_2_ which has a similar structure to graphene due to its conductivity and good electrical and chemical properties has been widely reported to be employed in numerous biosensors [43,44,45].

Hence, in this research, a new MoS_2_/Cu_2_O PN heterojunction structure, can reduce the predominant disadvantage of noise interference to the detection signal through different growth methods, material special characteristics, and the structure itself. This MoS_2_/Cu_2_O structure is applied to a sensor for cancer cells with different canceration levels. In this study, CVD is conducted for single-layer MoS_2_ to obtain large-area and uniform MoS_2_ films. Several measurement techniques, including Raman mapping, are used to check the number of layers. Atomic force microscopy (AFM) is employed for thickness analysis and scanning electron microscopy (SEM) is utilized to observe surface morphology. X-ray diffraction (XRD; Appendix A for XRD) is conducted to measure whether a signal appears, and transmission electron microscopy (TM; Appendix A) is performed to determine the direction of the crystal lattice. However, studies on the synthesis of semiconductor materials and nanostructures have also been carried out. During the research, Cu_2_O nanostructures have been successfully fabricated through electrochemical deposition. The major application of this heterostructure is to develop a low-cost and rapid way to detect lung cancer cell types in hydroplegia by fabricating a PEC biosensor that is highly sensitive to lung cancer cells significantly reducing the current busy procedures and longer waiting time for detection.

## 2. Materials and Methods

The glass substrate used in the experiment was indium tin oxide (ITO) for Cu_2_O electroplating. The resistance of the ITO surface was 7 Ω, and the thickness of the coating was 200 nm. All the chemicals used were all American Chemical Society (ACS) and guaranteed reagent (GR) to avoid the failure of results because of insufficient purity or residual impurities in the grown crystals, especially low-chlorine Sigma-Aldrich (Burlington, MA, USA) 98+% for sodium hydroxide because the solution of Cu_2_O deposition was susceptible to the influence of chloride ions to produce copper chloride precipitation.

### 2.1. Process and Steps of Electrochemical Cu_2_O Growth

#### 2.1.1. Substrate and Electrolysis Pretreatment

Cu_2_O was prepared through electrochemical deposition, and the substrate used was a 2 cm × 2 cm transparent conductive film made of ITO for Cu_2_O electroplating (Appendix A for the Material characteristics of Cu_2_O). P-type Cu_2_O has a small energy gap of 2.0 eV, which was a suitable conduction band [46,47]. Acetone, methanol, isopropanol, and deionized water were used to reduce the impurities, such as dust, oil stains, or residues, on the electrode surface after ITO plating. The adhesion of impurity contamination also improved the stability of film growth to obtain a higher-quality Cu_2_O film (Appendix A for the List of substrates, organic solvents, gases, and chemicals used to grow Cu_2_O and MoS_2_). In the cleaning steps, the electrode surface was first cleaned with acetone in an ultrasonic oscillator for 10 min to dissolve and remove surface oil stains or other organic impurities. After the specimen was shaken and washed with acetone, it was rinsed with methanol in an ultrasonic oscillator for 10 min to dissolve and remove residual copper contamination on the surface. Isopropanol was used for cleaning in an ultrasonic oscillator for 10 min to dissolve and remove surface oil stains or other organic impurities. The glass substrate was soaked in deionized water with a resistivity of 18.2 M W·cm (25 °C), and an ultrasonic oscillator was used for 10 min to remove residual organic solvents. The sample was taken out, and a high-pressure nitrogen gun was used to remove the moisture on the surface of the ITO glass substrate. The ITO glass substrate was baked in an oven at 100 °C for 30 min to remove the residual moisture on the substrate and placed in a clean moisture-proof box for later use. The cut graphite rods with a diameter of 1 cm and a length of 18 cm were subjected to electrochemical Cu_2_O deposition. The main purpose of cleaning the electrodes of the graphite rods was to reduce impurities, such as dust, oil stains, and residual powder, on the electrode surface during graphite rod production. External impurities contaminated Cu_2_O electrolytic deposition and improved film growth stability. For cleaning, the graphite rod was soaked in 1 M sodium hydroxide solution and washed with ultrasonic vibration for 30 min to remove grease and organic pollutants adsorbed on the surface. The surface of the graphite rod was ground with sandpaper and rinsed with deionized water during grinding to remove the powder remaining in the graphite rod manufacturing. The graphite rod was soaked in deionized water and shaken for 30 min with ultrasonic cleaning to reduce the residual impurities on the surface. The graphite rod was baked in an oven at 100 °C for 30 min to remove the surface moisture and placed in a moisture-proof box to complete the pretreatment of the graphite electrode.

#### 2.1.2. Preparation of Cu_2_O Film

A Cu_2_O thin film was prepared for electrochemical deposition to grow a highly uniform thin film on the ITO glass substrate (Appendix A for Cu_2_O Synthesis). The electrolyte was made of 0.4 M copper sulfate powder (CuSO_4_), 85% lactic acid, and 5 M sodium hydroxide (NaOH) prepared into a 1000 c.c. aqueous solution. The pH could be adjusted to 10, 11, and 12 through the amount of sodium hydroxide, which was measured in real time by a portable Starter300 pH meter (ST300). A DC power supply (MOTECH LPS505N) was used to power the two-pole electrochemical deposition system equipment in which the positive electrode was a graphite rod, and the negative electrode was connected to the ITO glass substrate (Appendix A for the electrochemical deposition system). The temperature of the growing electrolyte was controlled at 60 °C to ensure that the lactic acid works. The deposition time was fixed at 30 min to achieve high quality and uniformity. At the end of the film, the sample was washed with deionized water, and the surface moisture was removed with a high-pressure nitrogen gun for subsequent experiments.

#### 2.1.3. Cu_2_O Grinding

A plane grinder was widely used for the single-sided grinding and polishing of various materials, such as LED sapphire substrates, optical glass wafers, quartz wafers, silicon wafers, germanium wafers, molds, light guide plates, and optical skewer joints. The Cu_2_O thin film was processed by grinding to smoothen the Cu_2_O surface and facilitate the subsequent transfer of MoS_2_ while maintaining the integrity of the morphological characteristics of MoS_2_ and reducing the transfer failure rate. The rotation speed of the grinder was controlled at the minimum, and the time was set at 1 h. Considering Cu_2_O was a thin film at the micrometer scale, we chose the micro grade at the scale of a diamond polishing sheet.

#### 2.1.4. Experimental Materials and Drug Specifications of MoS_2_

The substrate used to grow MoS_2_ was silicon dioxide (SiO_2_) for chemical vapor deposition (CVD; Appendix A for obtaining MoS_2_ layers). The thickness of the chosen silicon wafer (100) crystal plane was 300 nm. The purity of sulfur powder (S) and molybdenum oxide (MoO_3_) powder was 99.98% and 99.95%, respectively, to avoid impurities that affect the CVD or the remaining impurities of grown single crystals. The quartz tubes and ceramic crucibles in the tubular thermal tubes were cleaned with aqua regia. The concentrations of nitric acid and hydrochloric acid were 37% and 68–69%, respectively, to avoid the adhesion of impurities from the previous growth during cleaning, which affects the experimental parameters and reproducibility of the next growth of monolayer MoS_2_.

### 2.2. Process and Steps of CVD-Grown MoS_2_

The majority of 2D material layer identification studies focus on film synthesis using mechanical stripping [48,49,50]. Most 2D material layer identification studies have focused on film synthesis through mechanical stripping [51]. In the present experiment, a 300 nm SiO_2_ silicon substrate was prepared and cleaned with ultrasonic vibration in acetone for 10 min to remove impurities and oil stains on the surface; deionized water was then used to remove the organic solvent acetone (Appendix A for CVD-grown MoS_2_).

After 10 min, the sample was taken out, and the moisture on the surface of the ITO glass substrate was removed with a high-pressure nitrogen gun. The method used to grow MoS_2_ was CVD [52,53,54,55,56,57]. The substrate was placed under specific pressure and temperature conditions and one or more precursors were chemically reacted on the surface of the substrate to produce a high-quality large-area thin film. The application of CVD in the preparation of single-layer TMDs starts with MoS_2_ growth. The inert gas used to grow MoS_2_ through CVD was argon. Afterward, 500 sccm of argon was used to clean the internal cavity and keep it in a clean environment. The heating rate was set to 20°C per min, the growth temperature was 650 °C, and the temperature was held for 30 min. Once the temperature was dropped to 400 °C, the lid was opened. The MoS_2_ structure was obtained when the temperature decreases to room temperature. The experimental process was shown in Figure 1. With molybdenum trioxide (MoO_3_) and sulfur powder (S) as precursors, a 2 cm × 2 cm silicon dioxide/silicon (SiO_2_/Si) substrate was placed on the crucible and sent into the furnace tube. The vaporization point of MoO_3_ was 650 °C [58,59,60]. The vaporization point of S was above 200 °C. Gas-phase MoO_3_ undergoes two chemical reactions in high-temperature environments to produce molybdenum oxide (MoO_3_-x) intermediates. The resulting molybdenum oxide intermediates diffuse to the substrate and vaporize to form a MoS_2_ film. The distance between the two crucibles was 46 cm. This long distance was to ensure that the vapor concentration gradient of sulfur was negligible on the substrate compared with that of the MoO_3_ concentration gradient because the distance between MoO_3_ and the substrate was small. CVD can effectively prepare single- and multi-layer MoS_2_. It can grow high-quality single-crystal materials and prepare uniformly distributed thin films on a large area, which was conducive to subsequent optoelectronic component manufacturing.

#### MoS_2_ Transfer Process

A spin coater was used to coat uniformly the photoresistant PMMA A5 on the substrate to transfer MoS_2_ after CVD growth on the Cu_2_O surface and left for the PMMA to dry. The substrate was covered with photoresistant liquid into 2 M NaOH as an etching solution for separating the substrate and PMMA A5. The removed photoresist was placed in DI water and blown dry with high-pressure nitrogen to prepare for the subsequent transfer. Then, PMMA A5 was placed on the original SiO_2_ substrate, which was cut to an appropriate size with a three-axis leveling table and placed on the surface of the Cu_2_O substrate. The temperature was set to 210° for about 30 s to make the PMMA adhere to the substrate. The sample was soaked in acetone for 30 min, and photoresistance was removed, leaving MoS_2_ to complete the entire transfer process (Figure 2).

## 3. Results

### 3.1. Growth Results of MoS_2_/Cu_2_O

Based on the above growth methods and measurement results, the CVD-grown MoS_2_ was transferred electrochemically to grown Cu_2_O (Appendix A for the results of Cu_2_O electrochemical growth under different parameters). Material and optical characteristics were analyzed: Raman mapping (Appendix A for micro-Raman Spectroscopy), SEM (Appendix A for scanning electron microscope), and SHG measurements (Appendix A for multiphoton excitation microscope).

#### 3.1.1. Cu_2_O Flattening Comparison

This study has used the Cu_2_O film grown by the electrochemical method as the template for MoS_2_ transfer. The SEM image is used to detect the success rate of the transfer, and the result is consistent with the Cu_2_O image without transfer in the SEM image, as shown in Figure 3a,b. The reason for the failure is that the Cu_2_O surface used for the electrochemical method is quite rough, and the rough surface causes the MoS_2_ transfer to be broken or unable to be transferred because of the uneven surface. Afterward, the Cu_2_O film was ground to smoothen the Cu_2_O surface and facilitate the subsequent transfer of MoS_2_ while maintaining the integrity of the morphological characteristics of MoS_2_ and reducing the transfer failure rate. Figure 4a shows the AFM data results of Cu_2_O before grinding. Under the following parameters, i.e., Z range of 740.16 nm, Rms of 97.496, and data scale of 0–800 nm (height), Figure 4b shows the results of Cu_2_O after surface polishing by grinding. Under the following parameters, i.e., Z range of 94.623 nm, Rms of 3.849 nm, and data scale of 0 50 nm (height), the top and flat views of the Cu_2_O film in the SEM image are shown in Figure 3c,d, respectively. The surface after grinding is smooth. According to the above data, grinding may be used to treat the Cu_2_O surface.

#### 3.1.2. MoS_2_ Transfer Result on Cu_2_O

After the Cu_2_O surface is polished by the abovementioned grinding method, the SEM image result shows that the surface is smoother than the original unpolished surface, and the surface roughness is reduced by 96% in the AFM measurement result (Appendix A for atomic force microscope). Then CVD-grown MoS_2_ film is transferred to the Cu_2_O surface, and SEM is performed (Figure 5). Although few irregularities are observed on the surface of the Cu_2_O, SEM analysis shows that MoS_2_ is transferred successfully, and the morphological characteristics of MoS_2_ are completely present on the surface of Cu_2_O.

#### 3.1.3. Analysis of MoS_2_/Cu_2_O Materials and Optical Properties

After the transfer of the CVD-grown MoS_2_ film onto the electrochemically grown Cu_2_O surface. Its material and optical properties are analyzed by employing OM (Appendix A for optical microscope), SHG images, SEM, and Raman and PL mapping. A multiphoton image is used to identify the signal display during the MoS_2_ transfer (Figure 6). The morphological characteristics of MoS_2_ grown by CVD are obvious. A core point exists in most triangles, and when this core point corresponds to a multiphoton image, the signal becomes stronger than the color of the triangles other than the core point. Figure 6b shows the multiphoton image of MoS_2_/Cu_2_O/ITO after the transfer. Although many blue irregularities in the image, they are not evenly dispersed to make it clear whether they are MoS_2_ triangles.

However, the comparison of the obvious bright spots in the image in Figure 6a suggests that a bright spot signal may be the core image after MoS_2_ transfer and because Cu_2_O is not completely flat like the SiO_2_ surface after being polished. Moreover, this bright spot at the center of MoS_2_ is due to the nucleation of MoS_2_. The growth parameters, such as carrier gas, growth temperature, precursors, substrates, and promoters can affect the nucleation and growth modes of MoS_2_ [61]. However, Ho Kwon Kim et al. has shown that the pre-exposure of growth substrates to alkali metal halides and the Mo precursor before the growth stage appears can suppress nucleation of MoS_2_ [62]. Therefore, the other blue part may be a multi-photon image of a mixture of MoS_2_ and Cu_2_O. As shown in Figure 5, the grown MoS_2_ has been successfully transferred to the Cu_2_O film. These two data can be combined, proving that MoS_2_ has been successfully transferred to the Cu_2_O film by grinding and transfer. Raman mapping, which is the best method to analyze the number of layers, has been used to analyze the image of the transferred MoS_2_/Cu_2_O sample [63,64,65]. In this study, a 532 nm band is used as a laser excitation light source. From Figure 7b we can observe that the blue-green area is the MoS_2_ film, and the red area is part of a Cu_2_O film. The 532 nm band was used as the laser excitation light source in this study. As shown in Figure 7a, the Raman shift of MoS_2_ has a peak at 384.3 before transfer while the Raman shift had a peak at 402.8 cm^−1^ after the transfer. The Raman shift of Cu_2_O before the transfer has a peak at 221.8 cm^−1^. After transferring MoS_2_ onto the Cu_2_O film, the Raman shift did reach peaks at 221.8, 384.3, and 402.8 cm^−1^. It can be clearly inferred that the Raman peak of Cu_2_O appearing at 221.8 cm^−1^ indicates that the surface of Cu_2_O was partially oxidized. Similarly, the Raman shifts of MoS_2_ at 384.3 and 402.8 cm^−1^ are in accordance with E_12g_ and A_1g_, respectively, and combined vibrations of two different phonons (A_1g_(M)−LA(M)), respectively, which indicates the existence of MoS_2_ structures [66,67,68,69]. In the Cu_2_O/MoS_2_ composite films, the diffraction peaks of Cu_2_O and MoS_2_ are still observed. The result shows that the grown MoS_2_ has a uniform film in the image and can replace the morphology. E_12g_ and A_1g_ vibration modes are the main signals for the judgment of MoS_2_. The two vibration modes are highly dependent on the thickness of MoS_2_. When the Δk value of the subtraction of E_12g_ and A_1g_ peaks is less than 20 cm^−1^, the analyzed MoS_2_ is regarded as a single-layer structure.

Figure 8a shows the OM image of MoS_2_/Cu_2_O, Figure 8b illustrates the 625-band image selected for the PL mapping image. The MoS_2_ spectrum consists of two peaks corresponding to the A1 and B1 excitons at 667 nm (1.86 eV) and 627 nm (1.97 eV), respectively. The measured PL for Cu_2_O grown by the electrochemical method was about 720–900 nm. In the measurement of the PN heterostructure material combination of MoS_2_/Cu_2_O, Raman measurement was first performed on the OM image. After the signal of MoS_2_ is found, PL measurement is carried out, and the combined structure is examined through the measurement results to show that the band range covers 625–900 nm. We found that a micro signal is generated at about 627 nm in the orange elliptical dashed circle, and the wavelength of 627 nm belongs to the B2 excitons of MoS_2_. The signal gradually increases when it approaches 670 nm. However, the Cu_2_O signal is not in this part because its light-emitting band starts to generate signals from 720 nm. Therefore, the Cu_2_O film may be too thick, and the single-layer MoS_2_ film thickness is only 0.7 nm, so the two heterostructure materials are combined in the wavelength range of 670–700 nm. The entire signal undergoes a red-shifted phenomenon because the Cu_2_O film signal is strong.

### 3.2. Photocurrent Response Analysis

This research uses the preparation of biosensors, and the photocurrent measurement system and impedance analysis system set up in the laboratory (see Appendix A for Analysis of Structural Characteristics of Photoelectrochemical Biosensors). The photocurrent measurement of the three lung cancer cells A549, H460 and H520 was carried out by the carrier transport mechanism of photogenerated charge, and the zigzags-toothed microelectrode was used to perform aggregation and impedance analysis of the cancer cells. In this study, we utilized two characteristic substances, glutathione (GSH) and glutathione disulfide (GSSG), to verify the detection mechanism of the developed biosensor. Generally, GSH in cells exists in two forms: 90% of GSH is present in the cytoplasm as reduced GSH, and 10% is present in the mitochondria. For healthy cells, the ratio of GSH to GSSG is greater than 10:1. However, when cells become cancerous, the ratio of GSH to GSSG decreases [70,71]. Because the majority carriers in N-type PEC are electrons and hence GSSG participating in PEC reaction is more significant. When irradiated, electron-hole pairs are generated, GSSG recombines with photo-generated electrons, and photo-generated holes are detected, forming a system of hole currents. As the degree of canceration becomes more severe, the concentration of GSSG contained in cancer cells increases, and the measured photocurrent also increases [72,73]. On the contrary, the majority of carriers in the P-type PEC biosensor are holes and hence GSH participates in the PEC reaction more significantly. After being illuminated, GSH recombines with photo-generated holes, photo-generated electrons are detected, and a system of electron flow is formed. The more severe the carcinogenesis of the structure, the smaller the GSH concentration and also a decrease of the photocurrent [74,75]. As shown in Figure 9 the photocurrent decreased with the increase of suspended cell density, which indicated that more GSH was involved in the PEC process. From Figure 10, it can be known that different cells have different amounts of change, there is a highly linear relationship between the photocurrent change and the number of cells, and the linear regression of the three cell types is as high as 99%. The slopes corresponding to the three cell lines A549, NCI-H460 and NCI-H520 were approximately 0.000452, 0.000749 and 0.000174, respectively. By measuring the magnitude of the photocurrent, we can calculate the regression curve equation, thereby identifying which cell of the three cells is. In the growth mechanism of Cu_2_O, as the thickness increases, the stress or dislocation will increase, resulting in a certain optimal thickness of the material. Previous studies also shows that the best growth thickness of Cu_2_O under the environment and parameters of our growth is 2 μm [76].

## 4. Discussion

When the MoS_2_ film has a single-layer structure, the A_1g_ signal is stronger than that of a multilayer film. If the MoS_2_ film has a multilayer structure, the E_12g_ signal is stronger than that of a single-layer film. The PL signal of MoS_2_ also decreases as the number of layers increases. Therefore, we can observe the contrast of the image by selecting the PL mapping images of 625 and 675 nm, corresponding to the color bar value. Single- and multi-layer MoS_2_ film signals are found in the selected 625-band image compared with much fewer signals in the 675-band image. When we grow the MoS_2_ film, single- and multi-layer structures form. In the selection of images for mapping, more multi-layer MoS_2_ films are covered. The PL measurement results in this study use a 532 nm laser as the excitation light source to measure the PL of a single-layer MoS_2_ film. The measurement results show that a strong signal is generated at the wavelength of 667 nm, and the energy is 1.84 eV after conversion. It is the energy gap of single-layer MoS_2_. However, from Gang Li et al., the energy band alignment of bulk MoS_2_ and single layer MoS_2_ was observed at 1.4 eV and 1.78 eV, respectively [29]. From our study, the single layer MoS_2_ was observed at 1.84 eV, close to the previous observations. The future scope of this project can be split into two aspects. First, after the two materials are combined, by using them as electric double-layer transistors their electrical properties can be tested. After confirming the positive state of the oxide trap. We can also try to bombard the element with high-energy particles to explore its characteristic influence and figure out if it affects the transformation of the mechanism. If the result of the electron beam bombardment affects the element and causes the characteristics to change, it can be checked again whether the electrical property changes from the positive oxide trap state to the negative interface trap state. Second, CVD can be performed to grow MoS_2_ in the 2H phase. MoS_2_ has three crystal structures: 2H (semiconductor characteristics), 1T (metallic characteristics), and 3R (semiconductor characteristics). The intensity difference can be detected in the PL mapping and electrostatic force microscope (EFM) images of 2H-MoS_2_ and 1T-MoS_2_. The future scope of this study is to collect a large amount of data to build a large-scale database by measuring the cell impedance and photocurrent in the patient’s pleural effusion. After analyzing the data through Artificial Intelligence (AI), a human-machine display interface will be designed which provides the types of cancer cells defined by the system, assisting physicians to determine the patient’s cancer status and provide appropriate treatment methods.

## 5. Conclusions

In this study, an electrochemical method was used to grow a Cu_2_O film with a uniform and superior crystal lattice. The micro-roughened surface structure of Cu_2_O was polished by grinding. Then, CVD-grown single crystal MoS_2_ was transferred to the Cu_2_O film. OM, SHG, SEM, Raman, and PL mapping measurements were used to analyze the combination of the two materials. Electrochemical methods were successfully used to achieve relatively good electrical properties. TEM showed that Cu_2_O is a single crystal structure, and SEM revealed that the surface should be processed by subsequent grinding methods. After grinding and polishing were performed, AFM and SEM were used to obtain the experimental results. MoS_2_ structure was grown through CVD, and E_12g_ and A_1g_ peaks were obtained via Raman analysis. The main luminescence peak at 667 nm was determined through PL analysis. The single layer of MoS_2_ was transferred to the polished Cu_2_O surface, which was confirmed by SEM and SHG results. Raman and PL mapping image analysis on MoS_2_/Cu_2_O indicated that the grown MoS_2_ had a uniform film. The combined structure was measured through PL measurement to verify that the band range covered 625–900 nm. A micro signal was found at the 627 nm wavelength. It belonged to the B2 excitons of MoS_2_ and tended to increase gradually as it approached 670 nm. The Cu_2_O film was too thick, whereas the thickness of the single-layer MoS_2_ film was only 0.7 nm. As such, the two heterostructure materials combined in the wavelength range of 670–700 nm. The strong signal of the Cu_2_O film itself led to a red shift in the entire signal. This study also successfully fabricated a low-cost PEC biosensor that is highly sensitive to lung cancer cells which is a rapid way to detect lung cancer cell types in hydroplegia. The photocurrent response is measured using the MoS_2_/Cu_2_O biochip material grown by electrochemical deposition. As the number of cancer cells measured increases, the content of oxidized GSSG also increases, and the measured photocurrent decreases accordingly. It is also possible to use dielectrophoresis to gather cancer cells to form a pearl string, measure unlabeled cancer cells, and use the slope of the admittance value to shape lung cancer cells. The linear regression curve is compared with the admittance value and the photocurrent measurement value to distinguish the types of cancer cells in the pleural effusion.

## Figures and Tables

**Figure 1 ijms-23-04745-f001:**
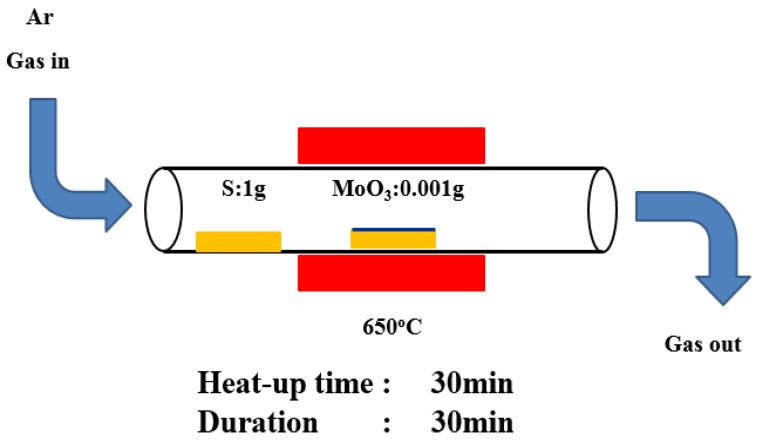
Schematic of a growing MoS_2_ film.

**Figure 2 ijms-23-04745-f002:**
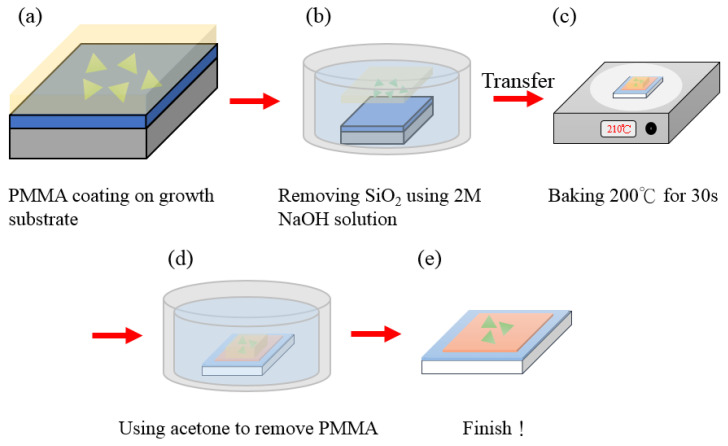
MoS_2_ transfer flow chart: (**a**) PMMA coating on growth, (**b**) Removing SiO2 using 2M NaOH solution, (**c**) Baking 200 °C for 30s, (**d**) Remove PMMA by acetone, (**e**) Complete the entire transfer process.

**Figure 3 ijms-23-04745-f003:**
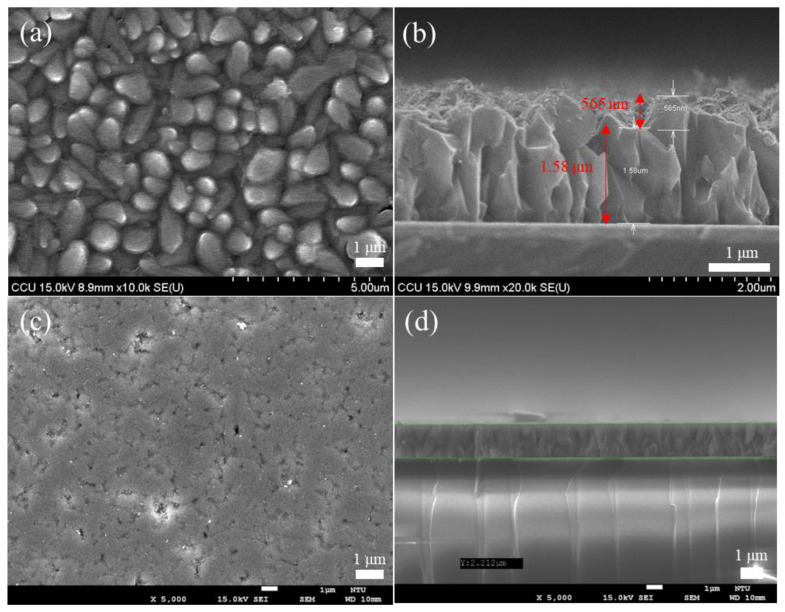
(**a**) Represents the front view of the Cu_2_O film before polishing, (**b**) represent the side view of the Cu_2_O film before polishing, (**c**) show the SEM images of the top view after Cu_2_O polishing, (**d**) show the SEM images of the front view after Cu_2_O polishing.

**Figure 4 ijms-23-04745-f004:**
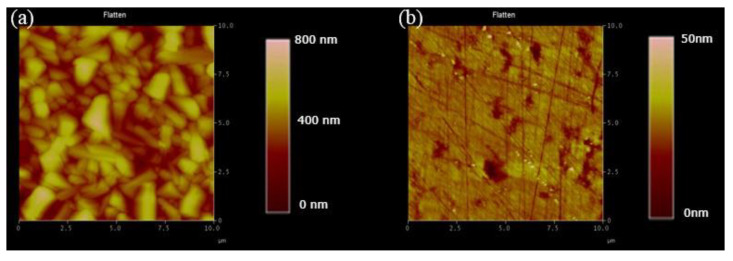
(**a**,**b**) AFM measurements of the Cu_2_O film before and after polishing.

**Figure 5 ijms-23-04745-f005:**
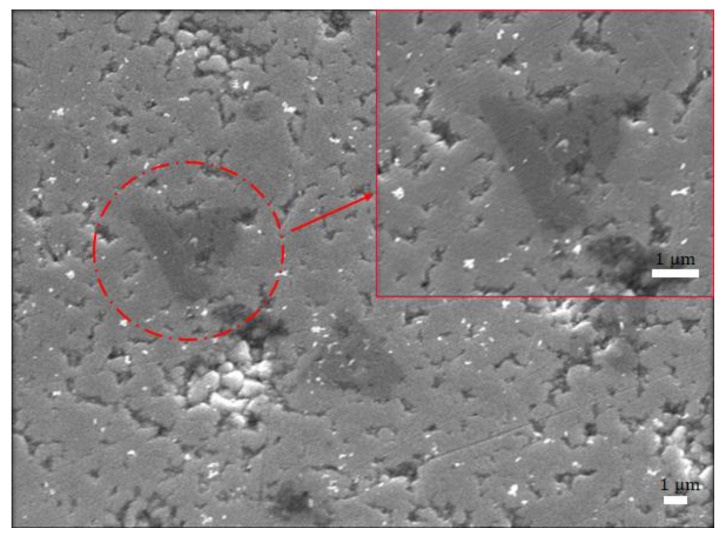
SEM image of MoS_2_ transferred to the Cu_2_O surface.

**Figure 6 ijms-23-04745-f006:**
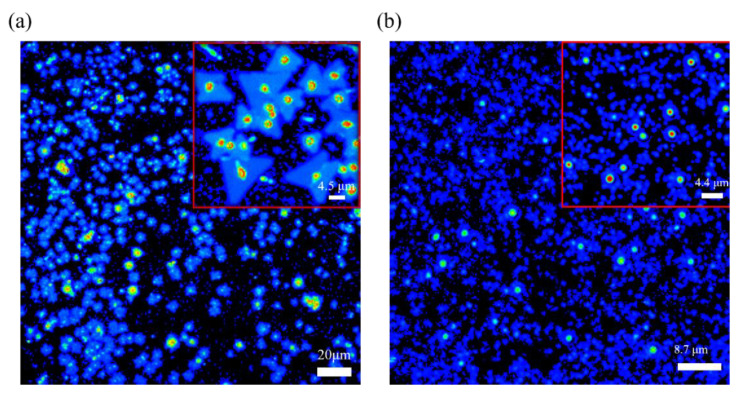
Multiphoton images of (**a**) MoS_2_/SiO_2_/Si and of (**b**) MoS_2_/Cu_2_O/ITO.

**Figure 7 ijms-23-04745-f007:**
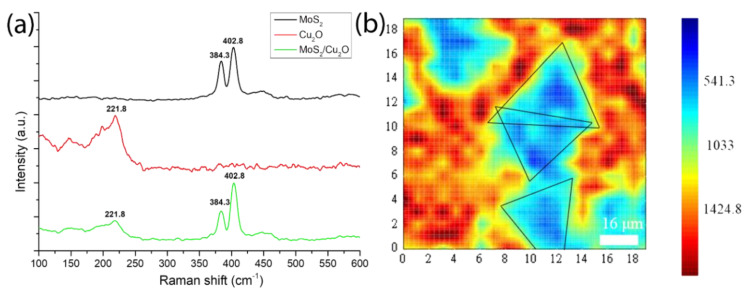
(**a**) Raman spectrum after MoS_2_ transfer to Cu_2_O surface and (**b**) Raman mapping of MoS_2_/Cu_2_O.

**Figure 8 ijms-23-04745-f008:**
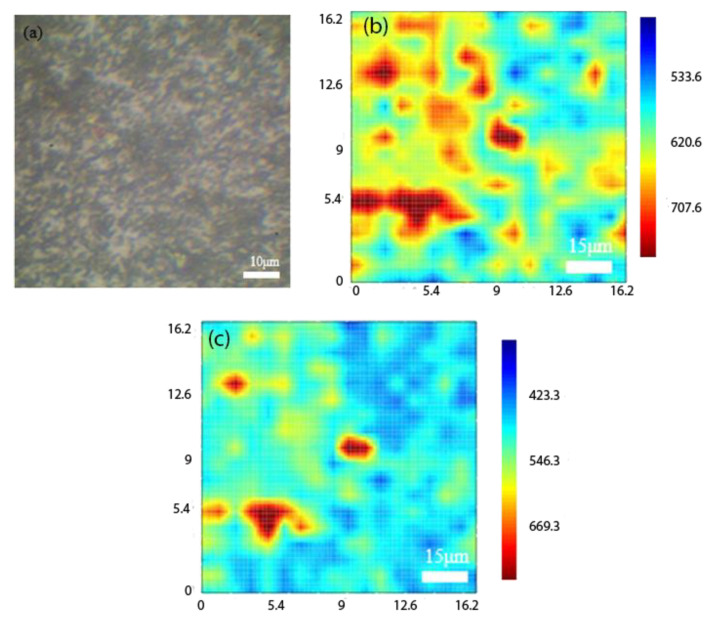
(**a**) OM image of MoS_2_/Cu_2_O, (**b**) 625 band for PL mapping image, (**c**) 675 band for PL mapping image.

**Figure 9 ijms-23-04745-f009:**
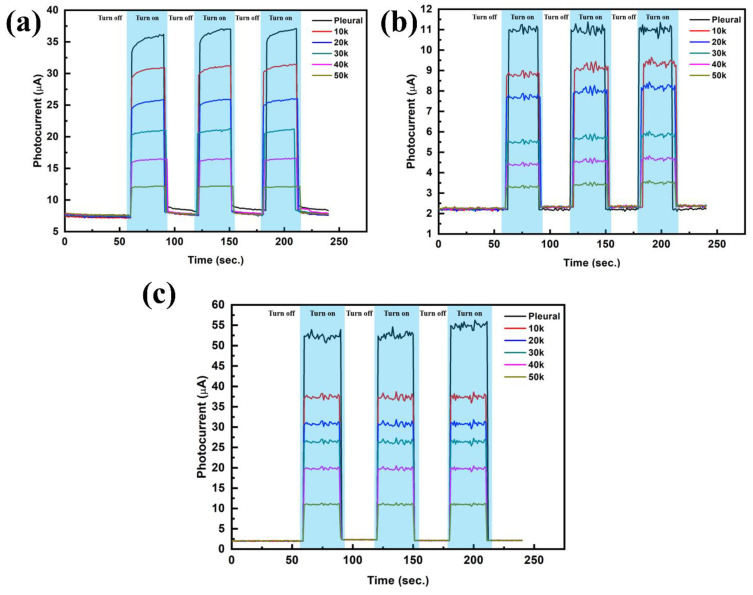
P. (**a**) Photocurrent responses of A549 under different cell numbers, (**b**) Photocurrent responses of NCI-H460 under different cell numbers and (**c**) Photocurrent responses of NCI-H520 under different cell numbers.

**Figure 10 ijms-23-04745-f010:**
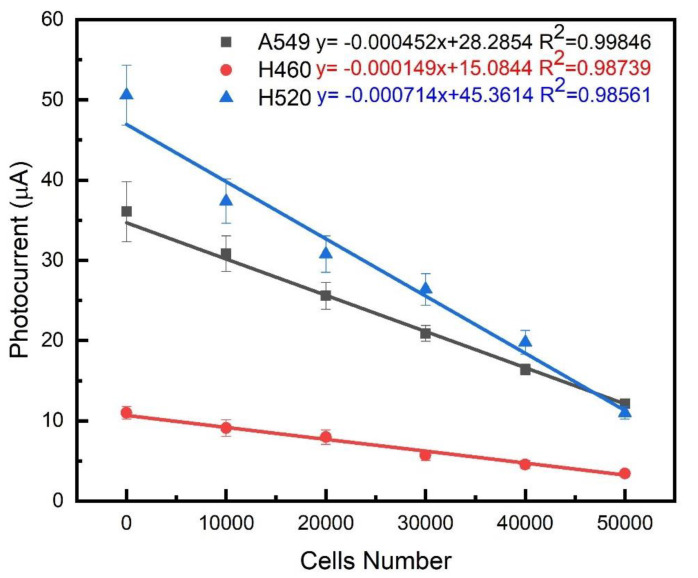
Linear relationship between photocurrent changes and cell number in three lung cancer cell lines on biosensing chips.

## Data Availability

Not Applicable.

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
