# Peer review of "Optical and Material Characteristics of MoS2/Cu2O Sensor for Detection of Lung Cancer Cell Types in Hydroplegia"

_ijms, 2022, doi:10.3390/ijms23094745_

Round 1

Reviewer 1 Report

In this work Wang et al., reports MoS2/Cu2O biosensor for the detection of lung cancer cells. Various characterization techniques such AFM, Raman and PL are used. The work is well presented however the reviewer has some suggestions, which are outlined below.

The abstract doesn’t give much information about the sensing part; therefore, it is suggested that the author should give some more details about sensing.

The objective of the paper in term of selecting MoS2/Cu2O is not clear. The authors should address it in clear way.

In the introduction section, the authors focused on the application Molybdenum disulfide (inorganic) based materials in sensing. Beside inorganic, organic materials are currently gaining attention in sensing applications also, therefore the author should put some details about organic material in sensing, and compare their properties with inorganic materials to broaden the introduction of the paper by referring to some relevant papers (doi.org/10.3389/fbioe.2019.00237, doi.org/10.3390/chemosensors9020030).

In line 92, it is mentioned that the resistance of ITO surface is 8W. The unit of resistance should be corrected.

As the thickness of the active layer can affect the sensing performance, therefore the authors should fabricate devices with different thicknesses to study the effect of thickness on senor performance.

Author Response

Reviewer 1:

  1. In this work Wang et al., reports MoS2/Cu2O biosensor for the detection of lung cancer cells. Various characterization techniques such AFM, Raman and PL are used. The work is well presented however the reviewer has some suggestions, which are outlined below.

Reply:

             The authors thank the reviewer for their positive comments. The authors have revised the manuscript based on the reviewer’s comments.

  1. The abstract doesn’t give much information about the sensing part; therefore, it is suggested that the author should give some more details about sensing.

Reply:

            The authors appreciate the suggestion of the reviewer. The authors have added the following lines in the abstract:

            Finally, the biosensor is used to detect lung cancer cell types in hydroplegia significantly reducing the current busy procedures and longer waiting time for detection. The results suggest that the fabricated sensor is highly sensitive to the change in the photocurrent with the number of each cells the linear regression of the three cell types is as high as 99%. By measuring the slope of the photocurrent, we can identify the type of cells and the number of cells.

  1. The objective of the paper in term of selecting MoS2/Cu2O is not clear. The authors should address it in clear way.

Reply:

            The authors appreciate the comment of the reviewer. Taking the suggestion of the reviewers the authors have added the following information in the introduction section:

            “On the other hand, in the recent years due to the excellent electrochemical properties of Cu2O, much research has been conducted towards its application as a sensor [39-42]. MoS2 which has a similar structure to graphene due to its conductivity and good electrical and chemical properties has been widely reported to be employed in numerous biosensors [43-45].

Hence, in this research, a new MoS2/Cu2O PN heterojunction structure, which can reduce the predominant disadvantage of noise interference to the detection signal through different growth methods, material special characteristics, and the structure itself. This MoS2/Cu2O structure is applied to a sensor for cancer cells with different canceration levels.”

  1. In the introduction section, the authors focused on the application Molybdenum disulfide (inorganic) based materials in sensing. Beside inorganic, organic materials are currently gaining attention in sensing applications also, therefore the author should put some details about organic material in sensing and compare their properties with inorganic materials to broaden the introduction of the paper by referring to some relevant papers (doi.org/10.3389/fbioe.2019.00237, doi.org/10.3390/chemosensors9020030).

Reply:

            The authors appreciate the comment of the reviewer. The authors have added the following lines:

“Apart from inorganic biosensors, organic materials are also getting attention in the last few years. Some biosensors using poly(methylene blue) (PMB), poly(alizarin yellow R), poly(azure A), poly(azure B), poly(azure C), poly(brilliant cresyl blue), and poly(thionine) has been widely studied and reported [31-37]. Even though these biosensors have good selectivity, their disadvantages include instability, cost, and a need of mediator in some cases [38].”

  1. In line 92, it is mentioned that the resistance of ITO surface is 8W. The unit of resistance should be corrected.

Reply:

            The authors would like to apologize for the mistake in the unit of the resistance. The authors have changed the unit from W to Ω.

  1. As the thickness of the active layer can affect the sensing performance, therefore the authors should fabricate devices with different thicknesses to study the effect of thickness on senor performance.

Reply:

It is correct to think about this question intuitively. Thicker active layers can absorb lighter and make the device more efficient. In the growth mechanism of Cu2O, as the thickness increases, the stress or dislocation will increase, resulting in a certain optimal thickness of the material. Previous studies also shows that the best growth thickness of Cu2O under the environment and parameters of our growth is 2 μm [76]. The authors have also added the aforementioned lines in the manuscript.

Reviewer 2 Report

The authors described procedures for preparation of the MoS2/Cu2O materials in order to develop a biosensor for detection of the lung cancer cells. The authors also used a powerful techniques in order to characterize prepared materials and did a tremendous job in characterization of materials and detailed presentation of results of the characterization. This is also the best and strongest part of the Manuscript. However, Introduction part provide sufficient information concerning MoS2 and Cu2O materials with their perspective and uses, but lacking with information about biosensors part. The authors must provide comprehensive and relevant literature of the uses of similar materials and techniques in determination of the various cancer cell. Also, Response analysis lacking of the interference studies and detailed optimization process.

In addition in the title of the Manuscript there is term "biosensors", although described sensor is not biosenor since biosensors are sensors in which the recognition system is based on biochemical or biological mechanisms.

Author Response

Reviewer 2:

  1. The authors described procedures for preparation of the MoS2/Cu2O materials in order to develop a biosensor for detection of the lung cancer cells. The authors also used a powerful techniques in order to characterize prepared materials and did a tremendous job in characterization of materials and detailed presentation of results of the characterization. This is also the best and strongest part of the Manuscript. However, Introduction part provide sufficient information concerning MoS2 and Cu2O materials with their perspective and uses but lacking with information about biosensors part. The authors must provide comprehensive and relevant literature of the uses of similar materials and techniques in determination of the various cancer cell. Also, Response analysis lacking the interference studies and detailed optimization process.

Reply:

             The authors appreciate the comment of the reviewer. Taking into the suggestion of the reviewer the authors have added the following lines in the introduction section:

            “Apart from inorganic biosensors, organic materials are also getting attention in the last few years. Some biosensors using poly(methylene blue) (PMB), poly(alizarin yellow R), poly(azure A), poly(azure B), poly(azure C), poly(brilliant cresyl blue), and poly(thionine) has been widely studied and reported [31-37]. Even though these biosensors have good selectivity, their disadvantages include instability, cost, and a need of mediator in some cases [38]. On the other hand, in the recent years due to the excellent electrochemical properties of Cu2O, much research has been conducted towards its application as a sensor [39-42]. MoS2 which has a similar structure to graphene due to its conductivity and good electrical and chemical properties has been widely reported to be employed in numerous biosensors [43-45].”

  1. In addition, in the title of the Manuscript there is term "biosensors", although described sensor is not biosensor since biosensors are sensors in which the recognition system is based on biochemical or biological mechanisms.

Reply:

            The authors apologize for the mistake. The authors have changed the title of the article from biosensors to sensors. Now the title of the article is, “Optical and Material Characteristics of MoS2/Cu2O sensor for Detection of Lung Cancer Cell Types in Hydroplegia”

Round 2

Reviewer 1 Report

The revised manuscript reflects the changes based on the feedback provided. So, I will believe it is suitable for publication in its current shape.